# Green Building Efficiency and Influencing Factors of Transportation Infrastructure in China: Based on Three-Stage Super-Efficiency SBM–DEA and Tobit Models

**Guijun Li** [1,2], **Xiaoteng Ma** [1,2] **and Yanqiu Song** [1,2,*]

1 School of Management Science and Engineering, Central University of Finance and Economics, Beijing 100081, China; ligj@cufe.edu.cn (G.L.); mmaxttt@163.com (X.M.)
2 Center for Global Economy and Sustainable Development, Central University of Finance and Economics, Beijing 100081, China
* Correspondence: songyq@cufe.edu.cn; Tel.: +86-13426335877

**Abstract:** As an important part of low-carbon ecological city construction, green building is also an objective requirement of sustainable development. Based on the green building panel data of 30 provincial administrative regions in China from 2010 to 2020, the super-efficiency SBM model combined with the three-stage DEA model was adopted to obtain the green building efficiency value that was closer to the real situation by excluding the influence of environmental factors and statistical noise. Green buildings in China have only been developing for just over ten years and are still in the initial stage of spatial aggregation in which transportation infrastructure plays an important role in scale effect. This manuscript focuses on analyzing the influence factors, intensity and direction of transportation infrastructure on green building efficiency. The results show that: (1) The agglomeration effect is obvious in the area of green buildings with high efficiency, but the radiation effect is not strong in low-efficiency area. (2) Municipal transportation infrastructure investment, road surface area, per capita number of stations and interregional traffic network density have a positive impact on green building efficiency, while freight volume has a negative impact.

**Keywords:** green buildings; transportation infrastructure; DEA model; Tobit model

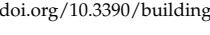



## 1. Introduction

In the face of energy supply shortage, energy conservation and emission reduction, environmentally sustainable development has received more and more attention and calls, causing the government and society to pay extensive attention to sustainable development. The construction industry, as one of the three basic industries with huge energy consumption [1], should also take the road of sustainable development. The development of green building has become a national and social development vision. As defined by the World Green Building Council, the green building is a building that has been designed, constructed or operated to reduce or eliminate negative environmental impacts and has a positive impact on the climate and natural environment. The United Nations Environment Programme has reported that through energy efficiency, fuel conversion and the use of renewable energy, green buildings have the potential to reduce carbon dioxide emissions by up to 84 billion tons by 2050 and save 50% or more energy to support limiting the global temperature rise by 2 degrees Celsius [2]. As one of the effective ways to reduce greenhouse gas emissions and solve the global energy crisis [3], green building has become a hot issue in academic research in recent years. Green buildings also provide social benefits, many of which are related to the health of those who work or live in green buildings. Studies have shown that green buildings may have a positive impact on public health through improving the indoor environmental quality (IEQ), which contributes to reductions in

perceived absenteeism and work hours affected by asthma, respiratory allergies, depression and stress and to self-reported improvements in productivity [4]. There is abundant evidence indicating that a better indoor environment leads to higher occupant satisfaction, productivity and well-being [5–8]. Although some research results show that the rating level and the product and version under which certification had been awarded, did not affect workplace satisfaction, there are also several studies providing some evidence that in the Occident (mainly the U.S. and the U.K.), no significant differences were found on occupant satisfaction between green and nongreen buildings and that in the Orient (mainly China and South Korea), green building occupants showed significantly higher satisfaction compared to nongreen building occupants [9]. In addition to the environmental benefits and social benefits, green buildings also provide economic benefits, such as saving on utility bills for the tenant or family by improving the efficiency of energy and water use, reducing the construction cost and improving the property value for building developers, thereby increasing the occupancy rates of the building owners and creating new employment opportunities and so on.

It cannot be ignored that no firm conclusions have been reached about the economic viability of green buildings. Cao Shen believes that by reasonably controlling the cost of the project, the cost of green building may be recovered in the whole life cycle of its construction [10]. Lynne Brouwer believes that the investment cost of green buildings is not proportional to the benefits obtained [11]. While going green is more likely to be seen as profitable from a building life cycle perspective, the economic viability from the perspective of developers and occupants remains unclear due to the cost issues arising from a variety of factors. There has been a widespread perception in the industry that "going green" is more expensive than traditional building methods [12–15]. After reviewing industrial reports and academic studies, a study concludes that the results reveal that the incremental costs for buildings certified as "green" range from 0.4% to 11%, and the authors note that such discrepancy in the results regarding economic viability is one major reason for the paradox of the very gradual diffusion of apparently cost-effective green buildings in most economies [16]. The vague debate over the economic viability of green buildings makes the issue still worthy of study.

It can be seen that there is an important link between the incremental cost of green buildings and transportation infrastructure. First of all, transportation costs are an important part of green building costs. Research shows that the key factor of the high cost of prefabricated components in China is the high proportion of transportation costs [17]. Therefore, the construction and operation of excellent transportation infrastructure will provide an external environment guarantee for the cost reduction of green buildings. There has been an overall trend toward reducing incremental green costs, as supply chains for green materials and equipment mature, and the industry becomes more skilled at delivering cost-effective green design and technology [18,19]. Secondly, from the perspective of China, the green building has only a decade of development history, which is still in the stage of relying on spatial agglomeration to produce scale effect to improve economic efficiency. At this stage, the transportation network environment plays a very important role, which will assist the industry to maximize the agglomeration effect and provide a guarantee for driving economic growth. And finally, transportation accessibility has always been an important evaluation indicator for green buildings [20]. In the Green Building Evaluation Standard (GB/T 50378-2019) of China, an evaluation target layer is set for green travel, which refers to the convenience of public transportation connections around the building. International green building evaluation systems, such as LEED BD + C(V4.0) in the United States and BREEAM in the United Kingdom, also set the evaluation layer of "location and transportation", which aims to encourage building projects to adopt various transportation modes to achieve the goal of reducing emissions and promoting public health. The transportation infrastructure system cannot only reduce freight and time costs, but also improve accessibility, thus improving the input–output efficiency of green buildings.

China is a country committed to sustainable development. In China, the energy consumption of buildings accounts for 46.7% of the total social energy consumption, and the energy demand of the construction industry accounts for 28% of the total energy consumption [21]. In this context, the development of green building is an effective solution to low-carbon ecological city construction. In 2013, The State Council of China issued *The Action Plan for Green Building Construction*, proposing that the proportion of green building area in new urban buildings will reach 70% by 2022. In 2019, more than 5 billion square meters of green buildings were built nationwide, accounting for 65 percent of new urban buildings in 2019. A total of 20,000 green building projects have been approved nationwide, with a floor area of more than 2.2 billion square meters. In 2020, the total area of green buildings in China exceeded 2.569 billion square meters. In 2020, the proportion of new green buildings in urban new civil buildings reached 77%, and the area of prefabricated buildings increased from 73 million square meters in 2015 to 630 million square meters [22]. By 2019, The Action Plan for Green Building Construction was revised to the third edition, with six indicators to build the green building evaluation system, respectively; the section with the outdoor environment, energy saving and energy use, water conservation and water resource utilization, saving material and material resource utilization, indoor environment quality, operation management (housing) and the whole life cycle of comprehensive performance (public buildings). The evaluation index system is divided into three stars, two stars and one star from high to low, and the specific indexes in each major index are divided into control item, general item and preferred item. Among them, the control items are the necessary terms for the evaluation of green building, and the preferred items mainly refer to the projects that are difficult to achieve and require higher indicators. Green buildings are classified into three grades according to the degree to which they meet the general and preferred criteria.

There has been a considerable amount of research in the field of green buildings. Some scholars have analyzed the factors influencing the development of green building, mainly using BP-Wings [23], fuzzy clustering [24] and analytic hierarchy process [25]. Some scholars analyzed the spatial–temporal distribution evolution characteristics of green building areas [26–28]. In addition, some scholars paid attention to the research on project risks and vulnerabilities [29–33], building scheme selection [34], carbon emission prediction [35], investment value index system construction [36] and other aspects. Data Envelopment Analysis (DEA), as an operational research and production boundary research method, has been applied in the field of green buildings research. Several studies adopt DEA models to study the benchmarking scheme selection [37–39]. And there also have been some studies set out to measure the efficiency in the construction industry using DEA models, but the scope of these studies is too broad to focus on the field of green buildings, such as building energy conservation policy and implementation of an efficiency evaluation [40], construction industry efficiency research [41], real estate enterprises and green productivity [42]. Or focus only on a single building type such as prefabricated building development efficiency [43] and security housing and construction efficiency [44]. It can be confirmed that the DEA method has been widely adopted in the field of green building efficiency research. However, it should still be noted that most of the existing literature on the efficiency of green buildings in China with the DEA method mainly focus on the regional scale [45–47] and do not expand to the national scale. The methods adopted are also the original DEA (first-stage DEA) or Slack-Based Model (SBM) DEA in which the influence of environmental factors and statistical noise is ignored. Based on previous research, this manuscript will also adopt the DEA method to measure the input–output efficiency of green buildings at the provincial level in China. Unlike previous studies, this study adopts the method of combining the three-stage DEA with super-efficiency SBM to eliminate environmental factors and statistical noise and strives to obtain the efficiency of green buildings more objectively and truly. Moreover, the research scope of this manuscript is the efficiency of green buildings nationwide in China from 2010 to 2020, which is more comprehensive and generalized than previous studies in both time and space. On the basis of efficiency studies, key influencing

factors will be identified with transport infrastructure as a breakthrough. The influencing factors will be analyzed from two paths of arrival costs and transportation costs to discuss the factors and directions of transportation infrastructure affecting green building.

## 2. Framework

The empirical analysis of this paper is divided into two parts. Firstly, the super-efficiency SBM model was combined with the three-stage DEA model in which the impact of slacks and external environmental factors on the efficiency measure was considered so as to evaluate the static input–output efficiency of green buildings in each province more accurately. Secondly, the Tobit model was adopted to carry out regression with green building efficiency values as independent variables and transportation infrastructure indicators as dependent variables. The results obtained by the Tobit model are the degree and direction of the impact of transportation infrastructure on the efficiency of green buildings. The above two-step analysis could provide the methods and paths to improve the efficiency of green buildings based on adjusting the layout of transportation infrastructure. The frame diagram is shown in Figure 1.

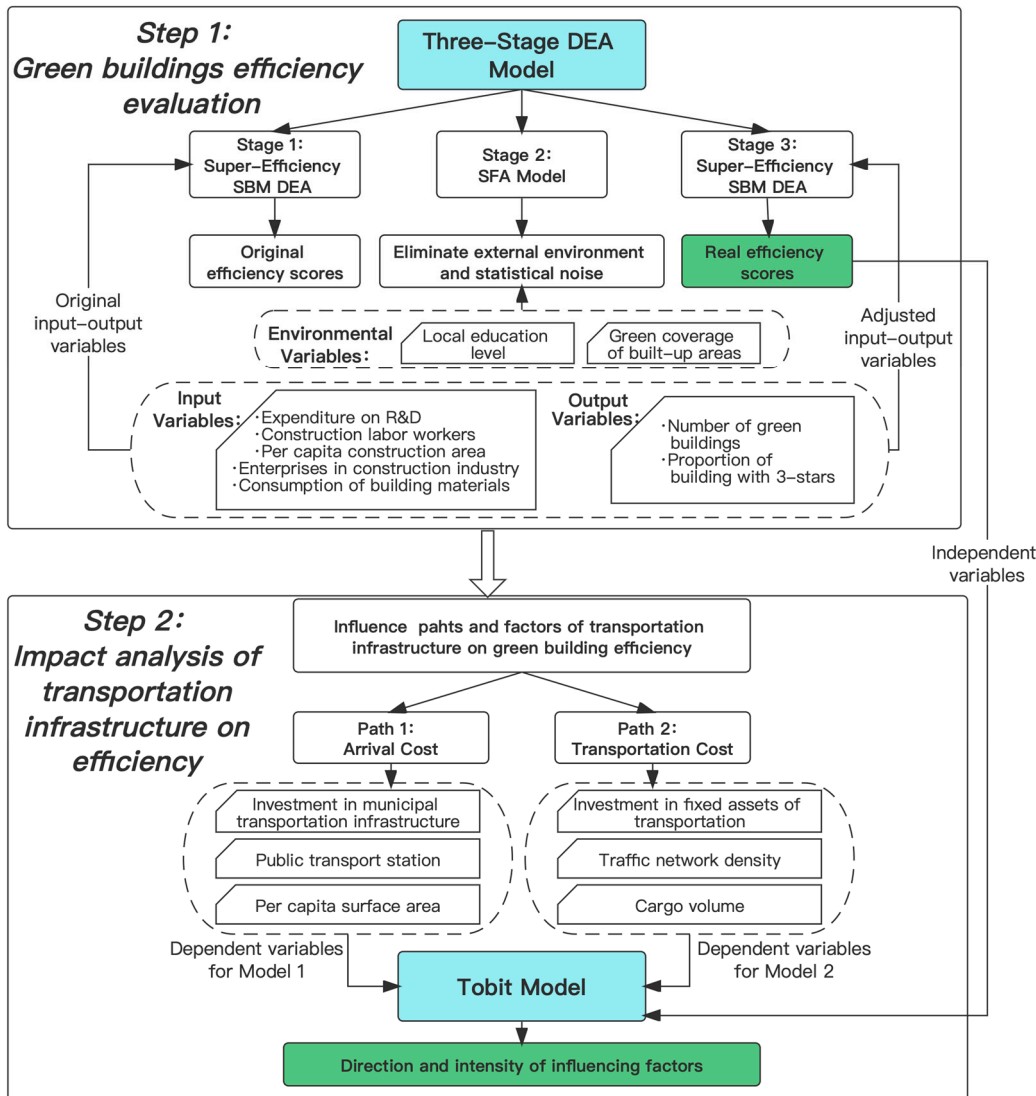

**Figure 1.** The methodology framework.

## 3. Methods and Data

### 3.1. Three-Stage Super-Efficiency SBM–DEA Model

According to Fried et al. [48], the three-stage DEA model could overcome the deficiency that the traditional DEA model and two-stage DEA model [49] could not estimate environmental variables and random errors. The biggest characteristic of this method is to adjust the input (or output) by using the information contained in the slack variables of the traditional DEA model so that all DMUs (decision making units) are adjusted to the assumed equivalent external environment. Then the traditional DEA model is used to recalculate the technical efficiency value of each DMU to eliminate the influence of external environmental factors so as to reflect the efficiency more truly. In order to make up for the defect that the SBM–DEA model cannot calculate all the efficiency values of every DMU, Tone [50] proposed the super-efficiency DEA model. That is, on the basis of the SBM model, the DMU with an efficiency of 1 is evaluated by using the super-efficiency SBM to obtain an efficiency value of more than 1. To maximize consistency with the actual production situation, this paper combines the super-efficiency SBM model with the three-stage DEA model. Taking the input-oriented DEA model as an example, the three-stage DEA super-efficiency SBM model divides the efficiency evaluation of the DMUs into three stages:

**Stage1: Super-Efficiency SBM Model**

Suppose $n$ decision-making units (DMUs) adopt m inputs, $x = x_1, x_2, \ldots, x_n \in R^{m \times n}$ to produce r outputs $y = y_1, y_2, \ldots, y_n \in R^{r \times n}$, and the data are positive. The model is expressed as follows:

$$\min \rho = \frac{1 - \left(\frac{1}{m}\right) \sum_{i=1}^{m} \left(\frac{w_i^-}{x_{ik}}\right)}{\frac{1}{r} \times \left(\sum_{s=1}^{r} \frac{w_s^+}{y_{sk}}\right)} \tag{1}$$

$$st. \quad \overline{x} = \sum_{j=1}^{n} x_{ij}\lambda_j + w_i^- \quad i = 1, \ldots, m$$

$$y_{sk} = \sum_{j=1}^{n} y_{sj}\lambda_j - w_i^+ \quad i = 1, \ldots, m$$

$$\lambda_j > 0 \quad j = 1, \ldots, n$$

$$w_i^- \geq 0 \quad i = 1, \ldots, m$$

$$w_s^+ \geq 0 \quad s = 1, \ldots, r$$

where matrices $x_{ij}$ and $y_{sj}$ represent $m$ inputs and $r$ outputs of $n$ DMU, respectively. Vectors $w_i^-$ and $w_i^+$ represent the excess value of input and the insufficient value of output, respectively. The DMU is SBM-efficient if and only if $\rho = 1$, i.e., $w_i^- = 0$ and $w_s^+ = 0$. In the discussion of super-efficient SBM, DMUs are defined as SBM-efficient. The super-efficiency SBM model, which can calculate the initial efficiency value, is expressed as follows:

$$\min \rho = \frac{1/m \sum_{i=1}^{m} \left(\frac{\overline{x}}{x_{ik}}\right)}{\frac{1}{r} \times \left(\sum_{s=1}^{r} \overline{y}\right)} \tag{2}$$

$$st. \quad \overline{x} \geq \sum_{j=1, \neq k}^{n} x_{ij}\lambda_j \quad s = 1, \ldots, m$$

$$\overline{y} \leq \sum_{j=1, \neq k}^{n} y_{sj}\lambda_j \quad s = 1, \ldots, r$$

$$\lambda_j > 0 \quad j = 1, \ldots, n$$

$$\overline{x} \geq x_k \quad s = 1, \ldots, m$$

$$\overline{y} \leq y_k \quad s = 1, \dots, r$$

where $\overline{x}$ and $\overline{y}$ represent the case with the least input and the largest output, i.e., the most efficient solution.

Referring to the literature related to the spatial evolution characteristics of green building area, site selection of green building, green building industry chain and other influencing factors of green building, the input and output variables are selected as follows.

**Input variables:**

Construction industry structure. The development of green buildings is influenced by the size of the construction industry. The economic benefits brought about by the expansion of the construction industry can attract the government's attention to the development of the construction industry, while the progress of the green building industry plays a substantial role in promoting energy conservation and emission reduction, which can promote the government's policy support to the whole industry to a certain extent. Construction enterprises concentrate a large amount of capital and professional and technical personnel to develop green building projects with their own conditions and obtain regional competitive advantages with better construction products. Therefore, the number of construction enterprises is taken as the input indicator. The data source is *China Construction Statistical Yearbook.*

Labor input. The development of green building is inseparable from the construction industry. Considering that the number of construction industry employees will affect the output of green building projects, the average number of construction labor and per capita construction area will be taken as input indicators. The data source is China Construction Statistical Yearbook.

Technology input. The design, construction and operation of green buildings are inseparable from advanced technology, and the full life cycle development of green buildings can be realized through effective methods such as innovation of architectural design technology and improvement of construction equipment level. The investment in technology is the driving force of the development of green building technology, which can be measured by the construction industry R&D expenditure. The data source is China Science and Technology Statistical Yearbook.

Construction consumables input. Compared with other energy consumption industries, the control of urban building consumables has more energy saving space, lower energy saving cost and obvious control effect, which is the key factor to accomplish the task of regional energy saving and emission reduction. At the same time, building material consumption and energy consumption are also an important part of the construction and operation costs of green buildings, so steel consumption is taken as the indicator to measure green building consumables. The data source is China Construction Statistical Yearbook.

**Output variables:**

The number of green buildings directly reflects the status quo of the green building industry, and the increase in its number can guide the development of the construction industry to the direction of greening. Therefore, the number of green building project signs is an important output indicator.

In addition, due to the different evaluation standards of green buildings with different stars, the higher the star level, the stronger the comprehensive ability of buildings in energy conservation and emission reduction, environmental quality and operation management. The proportion of green buildings with different stars of the total number of green buildings reflects the development quality of the green building industry, and the increasing proportion of green buildings with high stars is the direction of industrial structure optimization. According to the Green Building Evaluation Standard GB/ T50378-2019 issued by the Ministry of Housing and Urban-Rural Development of China, the three-star standard is the highest building standard in the green building evaluation system. According to the research of Wi et al. [51], the incremental costs of one-, two- and three-star green buildings are different, accounting for 2.7%, 6.2% and 9.3% of the overall construction costs, respectively. The difference in final function and cost of green buildings with different stars has a certain

impact on their input–output efficiency, so the proportion of three-star green buildings is selected as an indicator to measure the output quality of green buildings. The data source is Green Building Evaluation and Labeling Network.

Therefore, the number of green building evaluation and identification projects and the proportion of three-star green buildings were adopted as output variables.

Descriptive statistics of input and output variables are shown in Table 1.

**Table 1.** Descriptive statistics of input and output variables.

| | Variable | Unit | Observations | Mean | S.D. | Min | Max |
|---|---|---|---|---|---|---|---|
| input | Number of enterprises in construction industry | Enterprise | 300 | 2861 | 2138.7 | 104 | 11,000 |
| | Average number of construction labor workers | Person | 300 | 1,821,047 | 1,787,641 | 63,931 | 9,739,582 |
| | Per capita construction area | $m^2$/person | 300 | 196.7 | 77.8 | 55 | 595.9 |
| | Intramural expenditure on R&D of construction industry | 10,000 yuan | 300 | 150,722.8 | 374,952.7 | 668.8 | 3,386,904 |
| | Consumption of building materials (Steel) | 10,000 tons | 300 | $2.76 \times 10^7$ | $2.81 \times 10^7$ | 682,731 | $1.53 \times 10^8$ |
| output | Number of green building projects | Building | 300 | 17 | 31.7 | 0 | 287 |
| | Proportion of green buildings with 3 stars | % | 300 | 21.4 | 29.2 | 0 | 100 |

Note: S.D. stands for standard deviation, reflecting the degree of dispersion between individuals in the group.

**Stage2: Stochastic frontier analysis Model**

Due to the different external environment and statistical noise of each DMU in the first stage, the measured efficiency value failed to objectively reflect its real level. Therefore, based on the stochastic frontier analysis (SFA) proposed by C.P. Timmer [52], the slack variable is decomposed into a function containing three independent variables, namely environmental influence, statistical noise and management inefficiencies. According to the SFA regression analysis, the new outputs of each DMU in a homogeneous environment are calculated, which makes each DMU face the same external environment. The SFA regression model is expressed as follows:

$$S_{nk} = f^k(Z_n, \beta_k) + v_{nk} + u_{nk}, \, n = 1, 2, \dots, N; \, k = 1, 2, \dots, \quad (3)$$

where $s_{nk}$ represents the slack variables of the $k$ input of $n$ DMU, $f^k$ represents the function form corresponding to each of the slack variables, $z_n = (z_{1n}, z_{2n}, \dots, z_{mn})$ represents the value of $m$-environmental variables at $n$ DMU, $\beta_k$ represents the parameter vector of the $k$ input on $m$-dimensional environmental variables, $v_{nk} + u_{nk}$ is the compound residual term, $v_{nk}$ represents random error, $v_{nk} \sim N(o, \sigma_{kv}^2)$, $u_{nk}$ represents management inefficiency and follows the seminormal distribution and $u_{nk} \sim N^+\left(\mu^k, \sigma_{ku}^2\right)$, $u_{nk}$ and $v_{nk}$ are independent. $f^k(z_n, \beta_k) + v_{nk}$ represents the random possible margin boundary, and any margin beyond this boundary is affected by $u_{nk}$ and attributed to management inefficiency. Define $\gamma = \frac{\sigma_{ku}^2}{\sigma_{ku}^2 + \sigma_{kv}^2}$. When $\gamma$ approaches 1, management inefficiency is the main cause, and when $\gamma$ approaches 0, $u_{nk}$ can be removed from the model.

First, Fronter 4.1 is used to calculate the above parameters $\beta_k$, $\sigma_{ku}^2$, $\sigma_{kv}^2$. Then, the estimator $E[\mu_{nk}|v_{nk} + \mu_{nk}]$ of $\mu_{nk}$ is obtained by Jondrow et al. [53]. And the estimator of $v_{nk}$ could be obtained.

Using the estimation results of the SFA model, the input of each DMU is adjusted as follows:

$$x_{nk}^A = x_{nk} + [max\{z_n\hat{\beta}^k\} - z_n\hat{\beta}^k] + [max\{\hat{v_{nk}}\} - v_{nk}] \quad (4)$$

$$n = 1, 2, \cdots, N \, ; \, k = 1, 2, \cdots, K$$

The second term on the right of the equation means that the input of the *k* term of each DMU is adjusted to the amount needed to increase in the assumed situation that is most affected by environmental variables, that is, to make all the DMUs in the worst environment. The third term has a similar meaning: the amount of input needed to make DMUs at a maximum random disturbance. In this way, it is assumed that each DMU is in the same external environment and subjected to the same random impact, so the influence of these two factors on efficiency can be excluded.

**Environmental variables:**

Referring to the study of Simar and Wilson [54], environmental variables should meet the so-called "separation hypothesis", that is, factors, which have an impact on the efficiency of green buildings but are not within the subjective control of the sample, were selected.

According to Wu's [55] research, people with a higher education level usually have higher awareness of environmental protection. In general, since education level is generally proportional to residents' income level in most cases, residents with a higher education level are more likely to buy green products with higher prices [56]. Since the education level of the population in a region will affect the demand for green buildings, it will further affect the investment income expectation of green building developers, thus regulating the supply of green buildings. Therefore, the education level is selected as an environmental variable in this study, and the proportion of the population with a college degree or above was selected as a specific indicator.

Green building is a carrier of green life, and one of its core goals is to create a comfortable, healthy and sustainable lifestyle. To achieve the above goals, it is not enough to rely only on the development of green buildings. It requires joint efforts from various aspects to form a multiparty linkage, mutual promotion and complementary promotion mechanism. The green coverage rate of built-up areas can represent the coordination of regional green living goals to a certain extent, so it is selected as an environmental variable.

**Stage3: Adjusted DEA model**

The adjusted input data and the original output data are substituted into the BCC model to recalculate the efficiency values of each DMU. The information contained in the slack variables is used to separate the influence of external factors, and the efficiency value closer to the real situation is obtained.

*3.2. Tobit Regression Analysis Model*

The green building efficiency obtained through the super-efficiency SBM model is not only generated by the selected input and output indicators but also affected by other factors. In order to measure the direction and intensity of the main influencing factors, the two-stage method is gradually derived. The first step is to evaluate the green building efficiency of the DMU through the super-efficiency SBM model discussed above. The second step is to establish a regression model with the efficiency value obtained in the first step as the dependent variable and the influencing factors as the independent variables. The purpose is to judge the direction and intensity of the influence of the influencing factors on the efficiency of green buildings through the coefficient of the independent variables. Since this manuscript only examines the impact of transportation infrastructure on the efficiency of green buildings, other influencing factors are included into the control variables for analysis and investigation based on the traditional Tobit model. The Tobit regression model [57,58] cloud be written as:

$$Y = \begin{cases} Y^* = \alpha + \beta_1 factor_p + \beta_2 X + \varepsilon & Y^* > a \\ 0 & Y^* \leq a \end{cases} \tag{5}$$

In Formula (5), *Y* is the truncated dependent variable vector, $factor_p$ is the independent variables vector in different paths to impact the efficiency of green buildings, *X* is the control variable, *α* is the intercept term vector, *β* is regression parameter vector, and *ε* is the disturbance term, $\varepsilon \sim N(o, \sigma^2)$.

In the Tobit regression model, the efficiency values of green building as dependent variables are all positive values, which belong to truncated discrete distribution data. Therefore, truncation point a was set to 0.

In addition, it is biased to estimate model parameters by the ordinary least squares method (OLS), so maximum likelihood method (ML) was needed to estimate model parameters.

According to the analysis of this manuscript, the impact of transportation infrastructure on the efficiency of green buildings is mainly in two ways. One is to affect the arrival costs and the other is the transportation costs. The theoretical analysis of the impact will be detailed in the Tobit model regression results and discussion section. In the arrival cost model, fixed asset investment in municipal transportation infrastructure, road surface area per capita and the number of stations per capita are considered to be the core factors affecting the efficiency of green buildings. In the transportation cost model, the density of the regional transportation network and the volume of cargo transportation are the main factors.

### 3.3. Research Scope and Data Sources

Due to different certification standards and availability of data, the research object of this paper is the green building efficiency of 30 provinces excluding Hong Kong, Macao, Taiwan and Tibet. In this manuscript, 2700 data points including 5 input variables, 2 output variables and 2 environment variables were collected in the DEA model, and 2700 data points including 5 core variables and 4 control variables were collected in the Tobit model.

Although *The Green Building Evaluation Standard* has been revised three times, the standard of green building stars has not changed much and will not have a significant impact on the data results. The Green Building Evaluation and Labeling Network, organized by the Ministry of Housing and Urban-Rural Development's Center for Science, Technology and Industrial Development, has compiled data on green buildings since 2008. Considering that the green building in China was still in its infancy from 2008 to 2009, the spatial layout of it during this period was too sparse, which would affect the balance of the overall data. In order to ensure the value of the study and the integrity of the data, this manuscript select the data of green buildings from 2010 to 2020 for analysis and matched other relevant data with green building data as a standard. It should be noted that The Green Building Evaluation and Labeling Network did not publish the number of green building project signs in 2017, and the data of 2017 could not be obtained through other channels, so all samples did not contain the data of 2017. Fortunately, when the efficiency value was calculated by the DEA method, all the data were only compared and analyzed with the data of different provinces in the current year, instead of cross-year analysis. Therefore, the loss of data for a single year did not have any effect on the overall efficiency value.

## 4. Results

### 4.1. Three-Stage SBM–DEA Model

In general, the SBM–DEA model can be divided into two categories: input-oriented model and output-oriented model. An appropriate model is useful to guide production and operation activities. Since green building project identification is assessed by the Ministry of Housing and Urban-Rural Development of China, which means that output cannot be adjusted, our research aims to improve efficiency by adjusting factors affecting input. Therefore, the input-oriented DEA model was adopted in this paper to analyze the collected data through MaxDEA software.

Since the efficiency value of the DEA in the first stage may be affected by environmental factors and statistical noise, we needed to place different provinces under the same environmental level to adjust the efficiency value. The results of the SFA analysis are shown in Table 2.

**Table 2.** Regression results of SFA model in stage 2.

| | Construction Enterprises | Construction Labor | Construction Area | Expenditure on R&D | Building Materials |
|---|---|---|---|---|---|
| Afforestation coverage rate | −0.79 | −5275.84 *** | −0.79 * | 1681.64 *** | −49,592.57 *** |
| Education level of population | −25.69 | 18,796.52 *** | −2.61 * | 469.70 | 271,120.10 *** |
| Sigma-squared | $8.03 \times 10^6$ | $4.40 \times 10^{12}$ | $8.79 \times 10^4$ | $1.03 \times 10^{12}$ | $1.21 \times 10^{15}$ |
| gamma | 0.70 | 0.77 | 0.77 | 0.93 | 0.60 |

Note: *** $p < 0.001$, ** $p < 0.01$ and * $p < 0.5$.

Most of the environmental variables passed the significance level test, indicating that there are differences in comprehensive technical efficiency before and after, and it is appropriate to use the SFA model to analyze. Moreover, technical inefficiency has a great influence on the generation of input balance variables, while the influence of random error factors is small. When the regression coefficient is positive, it means that increasing the environmental variable will increase the amount of input slack, resulting in an increase in waste and a decrease in overall efficiency. On the contrary, when the regression coefficient is negative, this environmental variable is beneficial to reduce the amount of input slack, reduce the generation of waste and improve the overall efficiency. According to the regression results, we can see that:

(1) The impact of the green coverage rate on the average number of workers, the average construction area and the consumption of building materials in the built-up area is significantly negative, indicating that increasing the green coverage rate shows a positive increase in the demand for both labor and building material consumption in the construction industry, so it has a significant positive effect on the improvement of green building efficiency. This regression result also confirms the positive synergy and mutual promotion between the two mentioned above.

(2) The population by education level of construction and building materials, labor regression coefficient is positive, the possible reason for this is that the population affected by the higher education level, labor force, the easier to the more advanced industrial structure, and construction as the second industry, the less labor, at the same time, building material utilization and the usage of new materials are not highly educated population dividend. Therefore, it is not conducive to improving the efficiency of green buildings. The regression coefficient between the education level of the population and the per capita construction area is negative, indicating that with the improvement of education level, labor productivity generally improves, so the per capita construction area increases accordingly.

The efficiency score obtained by DEA model is technical efficiency (TE), which can be decomposed into the product of scale efficiency (SE) and pure technical efficiency (PTE). The TE = SE × PTE. Wherein, TE represents the ability to achieve the maximum output or the minimum input under the given input; SE represents the extent of economy of scale compared with the effective point of scale; PTE represents the efficiency of removing the factor of size.

The technical efficiency of stage 1 and stage 3 was decomposed, and the comparison of the average efficiency of each province before and after adjustment was obtained as shown in Figure 2.

After the adjustment of the SFA model, the average efficiency scores are improved, and the SE always plays a leading role in the TE, which is consistent with the trend of the TE.

The years of 2010, 2015 and 2020 were extracted from the results obtained by the DEA in the third stage for analysis, and the Arcgis software was used to draw the efficiency distribution map, as shown in Figure 3.

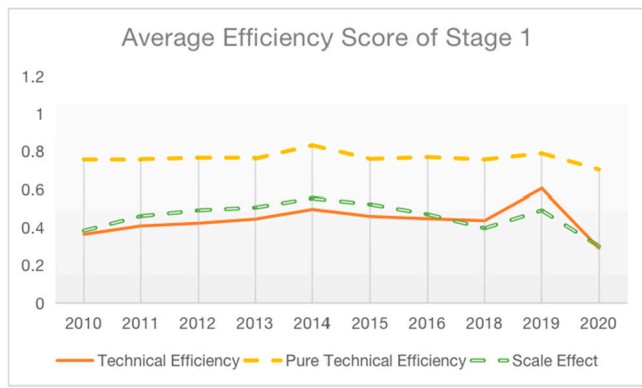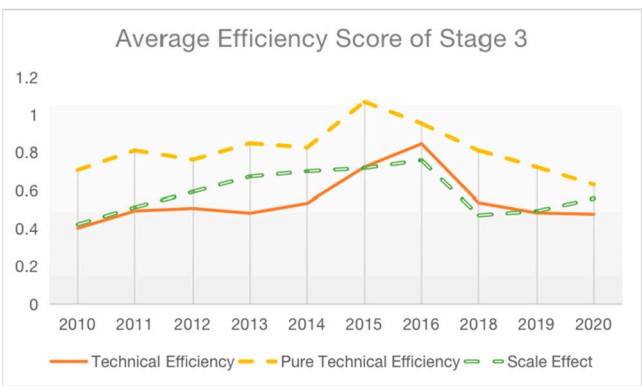

**Figure 2.** Average efficiency score of DEA model stage1 and stage3.

Figure 3a–c respectively show the TE (technical efficiency) distribution of green buildings in 2010, 2015 and 2020. As can be seen from the figure, the efficiency distribution of green buildings shows obvious spatial heterogeneity. The areas with high efficiency in the early stage are mainly concentrated in the eastern and southern coastal areas, which is consistent with the spatial distribution of green building projects and areas [59]. The high-efficiency regions tend to converge from the east and west to the central provinces, while the low-efficiency regions remain concentrated in the northeast and southwest provinces. In the calculation period, the average efficiency of green buildings in the eastern provinces such as Zhejiang province and Guangdong province remains highly effective continuously, while other provinces in the same region have remained in a state of low efficiency for years. By comprehensive comparison of location conditions, it is not difficult to find that efficient and effective areas are geographically adjacent to each other, which can maximize the intensive utilization of resources and environment in the development process of green buildings, forming an obvious internal cluster effect. However, the provinces with high green building efficiency are not sufficiently radiating and motivating to the surrounding areas. Instead of taking advantages of the benefits brought by the momentum of prosperity in efficient provinces, the provinces around have been inefficient throughout the measurement period. The efficiency gap is obvious in 2010 and 2015, and is not improved until around 2020.

Figure 3d–f show the distribution of PTE (pure technical efficiency) of green buildings. The distribution and tendency of PTE are different from that of TE. High-efficiency provinces are always distributed in northwest China, and the efficiency level of neighboring regions is also high, indicating that the effect of the cluster of PTE in northwest China plays a role. At the same time, PTE gradually converges from coastal and frontier areas to the center over time. Low efficiency provinces shifted from being concentrated in the central region to spreading to the southeast. Specifically, most provinces in the east, except Jiangsu province, Guangdong province and Hunan province, were in a depressed state of inefficiency, indicating that the radiation effect in the central and eastern regions was poor.

Figure 3g–i reflect the distribution and variation of SE (scale efficiency) of green buildings. The distribution trend of SE is basically consistent with that of TE, indicating that compared with PTE, SE dominates the overall situation of green building efficiency. This is consistent with the industrial development law of green buildings, that is, in the initial development stage, green buildings improve their efficiency relying on the scale effect, which generating by well-developed transportation infrastructure. But after a period of development process, knowledge spillover and scale economy gradually stabilize and come into play, and technology can occupy a dominant position in industrial efficiency.

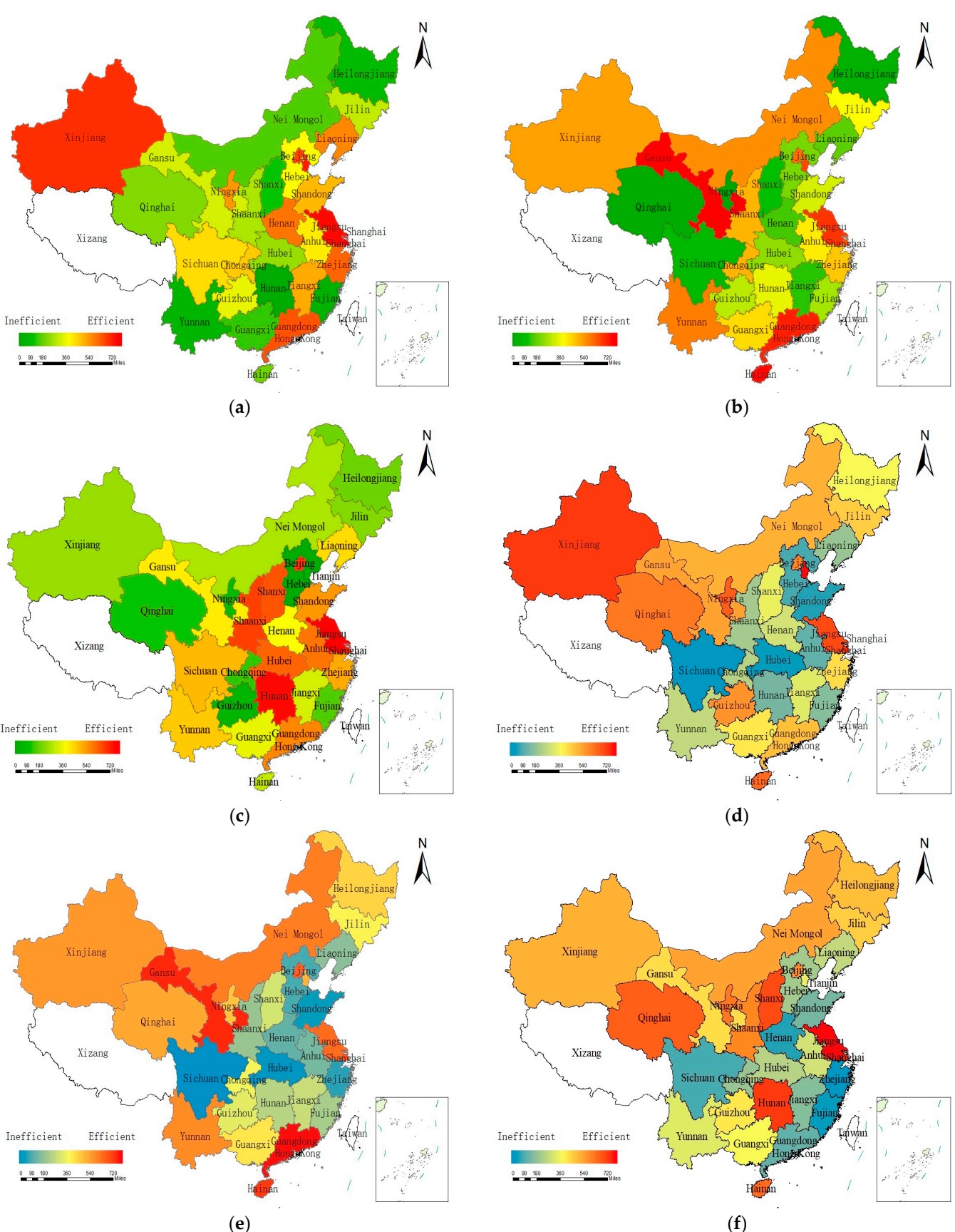

**Figure 3.** *Cont.*

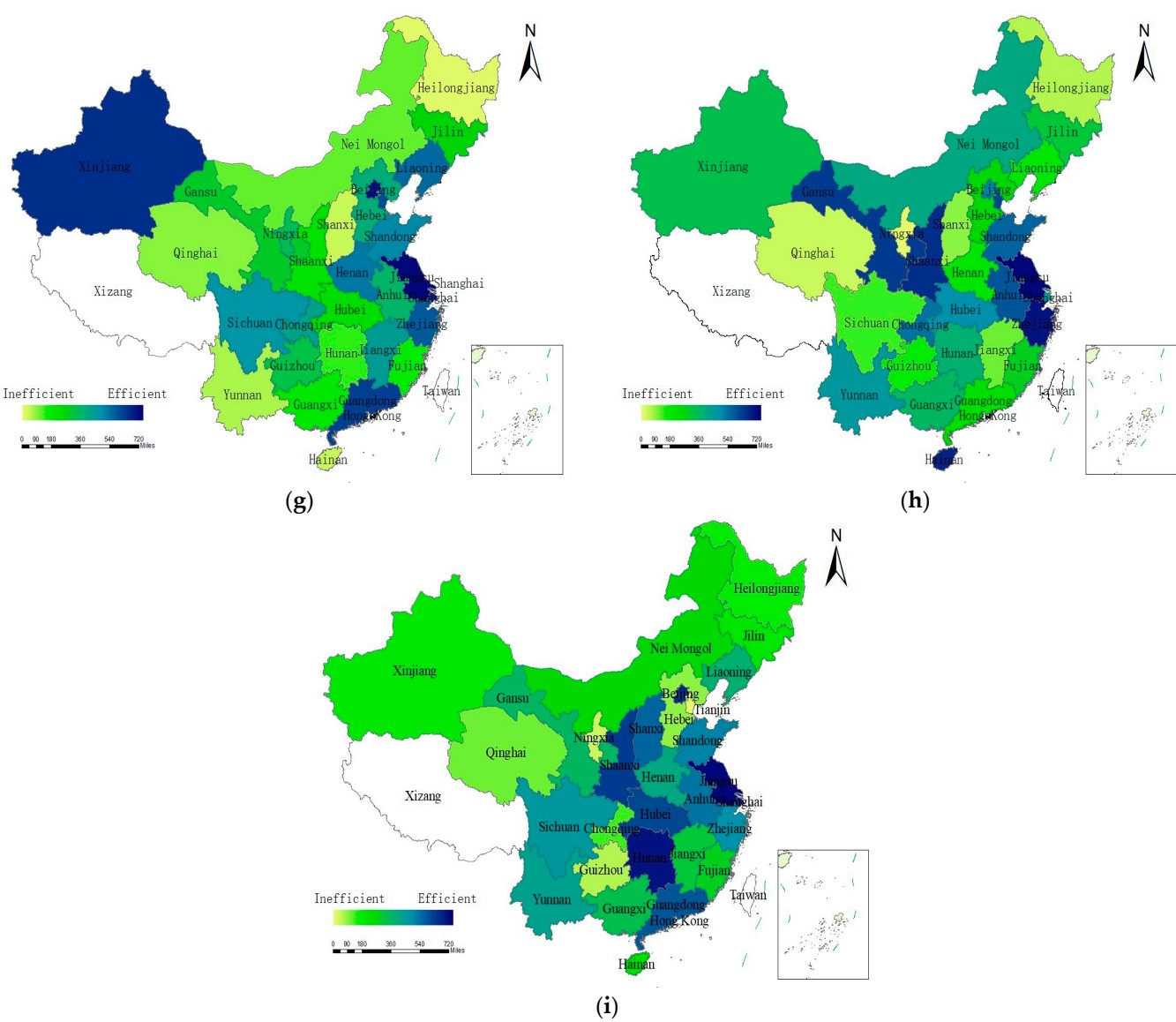

**Figure 3.** Green buildings efficiency and decomposition in China. (**a**) 2010 TE; (**b**) 2015 TE; (**c**) 2020 TE; (**d**) 2010 PTE; (**e**) 2015 PTE; (**f**) 2020 PTE; (**g**) 2010 SE; (**h**) 2015 SE; (**i**) 2020 SE. Note: Data missing in white area are not included in the research scope.

### 4.2. Tobit Model

As the input–output efficiency results obtained by the three-stage super-efficiency SBM–DEA model are typical censored variables, we adopted the Tobit model to analyze the transportation infrastructure factors affecting the efficiency of green buildings.

There are two main ways for the transportation infrastructure to affect the efficiency of green buildings: one is to reduce the arrival cost to improve accessibility, and the other is to reduce the transportation cost to reduce the construction investment of buildings. The difference between these two paths is that the former is influenced by the road network within the city, while the latter is mainly influenced by the transportation infrastructure between cities and even within the larger administrative divisions.

Specifically, the arrival cost is an important objective of green building design and site selection. In the Green Building Evaluation Standard (GB/T 50378-2019), the indicator for green passage is set as one of the scoring points of the evaluation target floor, which mainly refers to the convenience of public transportation connections around the building. Its scoring points require that there should be a public transport station or shuttle within 500 m of the pedestrian entrance of the site. In addition, in the operation stage of green buildings,

public transportation also plays a role that cannot be ignored. Yang (2020) believes that shared public transportation can improve the adaptability and comfort of green buildings in the outer suburbs of cities and expand the evaluation range of green buildings by three times [60].

The impact that taking transportation cost as the path mainly occurs in the construction stage of green building. As a secondary industry in China, the construction industry has formed a distinct division of labor in different locations after long-term development, so it is highly dependent on the transportation industry. The raw materials of green buildings need to be transported by railways, roads and waterways, and Xiong (2021) believes that green buildings have become an important direction in the reform of the construction industry, but the high transportation cost of prefabricated components has become a key factor hindering the investment and promotion of the prefabricated building production base and related green building [17].

Based on the existing literature and assessment standard for green building, this paper constructed two Tobit models to investigate the influencing factors of efficiency from arrival cost and transportation cost, respectively, as follows:

**Model 1**

$$GBE_{it}^* = \beta_0 + \beta_1 investment_{it} + \beta_2 surface_{it} + \beta_3 station_{it} + \beta_4 X_{it} + \varepsilon_{it} \tag{6}$$

The formula above is the arrival cost model in which $GBE_{it}^*$ represents the green building efficiency of province $i$ in year $t$, $investment_{it}$ represents for fixed asset investment in municipal transportation infrastructure, $surface_{it}$ represents the surface area of road per capita, $station_{it}$ represents the number of public transport stations per 10,000 people, $X_{it}$ represents a collection of control variables (which includes other factors that may influence green building, such as the level of local economic development, construction structure, technology and resource input), and $\varepsilon_{it}$ is the random error term. The data used in Model 1 are from China Urban Construction Statistical Yearbook 2010–2021. Table 3 shows the regression results of Model 1.

It can be seen from the regression results of Table 3 that in the arrival cost model, transportation infrastructure indicators have a positive effect on the efficiency of green buildings, and almost all of them have passed the significance test at the level of 5%, indicates that the three factors have a great positive impact on the development efficiency of green buildings.

(1) The regression result of fixed asset investment in municipal transportation infrastructure is significantly positive because the increase in fixed asset investment could comprehensively improve the overall situation of transportation infrastructure [61], and although green building is only one of the beneficiaries, its arrival cost is still reduced objectively.

(2) Road surface area per capita is usually used to measure the development of road infrastructure, which is also the best indicator to reflect the congestion of a city [62]. The increase in road surface area per capita leads to smoother traffic, reduced commuting time and cost and thus reduced arrival cost of green buildings.

(3) The number of stations per capita represents the flexibility and accessibility of transportation, and the increase in the number usually implies the increase in the mileage and carrying capacity of public transportation [63,64], which also means that more alternative modes of transportation may be arranged around green buildings. Increased accessibility reduces the cost of arrival, thus improving the overall efficiency of green buildings.

(4) When the impacts of transportation infrastructure on PTE and SE are further investigated, it can be found that the explanatory variables have a higher impact on PTE than on SE. The reason why fixed asset investment in municipal transportation infrastructure is not significant to SE may be that infrastructure investment in many provinces in China has shifted from pursuing economies of scale to improving technological content and unit efficiency [65,66]. It could also explain that the coefficients of the other two influencing factors of PTE are higher than those of SE.

**Table 3.** The regression results of transportation infrastructure factors affecting green building efficiency in arrival cost model.

| Variable | Regression Results | | |
|---|---|---|---|
| | **TE** | **PTE** | **SE** |
| Core Variables | | | |
| Investment | 0.22 * | 0.011 | 0.0869 |
| | (2.35) | (2.98) | (1.63) |
| Surface | 0.0314 ** | 0.0143 * | 0.0116 |
| | (2.73) | (2.05) | (1.77) |
| Station | 0.0027 ** | 0.0016 ** | 0.0017 *** |
| | (3.14) | (2.98) | (3.49) |
| Control variables | | | |
| GDP | −0.458 *** | −0.434 *** | −0.092 |
| | (−3.37) | (−5.25) | (−1.19) |
| Enterprise | 0.0001 | 0.00003 | 0.00004 |
| | (1.73) | (1.12) | (1.4) |
| R&D | $6.1 \times 10^{-7}$ *** | $4.03 \times 10^{-8}$ | $3.05 \times 10^{-7}$ *** |
| | (3.97) | (0.43) | (3.49) |
| Employee | 0.0005 | 0.0008 | 0.0009 * |
| | (0.73) | (1.89) | (2.00) |
| Constant | 0.5786 | 2.8908 *** | −0.5689 |
| | (0.54) | (4.43) | (−0.93) |
| Observations | 300 | 300 | 300 |
| Pseudo $R^2$ | 0.1 | 0.11 | 0.21 |

Note: t-statistics in parenthesis. *** $p < 0.001$, ** $p < 0.01$, * $p < 0.5$.

A similar method was used to analyze the impact from the perspective of transportation cost, and the Tobit regression model is established as follows:

**Model 2**

$$GBE_{it}^* = \beta_0 + \beta_1 density_{it} + \beta_2 volume_{it} + \beta_3 X_{it} + \varepsilon_{it} \qquad (7)$$

Formula (7) is the transportation cost model in which $GBE_{it}^*$ represents the green building efficiency of province $i$ in year $t$, $density_{it}$ represents the transport network density, $volume_{itj}$ represents the freight volume, $X_{it}$ represents a collection of control variables, and $\varepsilon_{it}$ is the random error term. The data used in Model 2 are from *China Statistical Yearbook 2010–2021*. The regression results are as follows:

The regression results of Table 4 show that the impact of the freight volume on the TE of green buildings is significantly negative, while the traffic network density is positive but not significantly.

(1) The volume of cargo transportation reflects the flow of factors. If there is a large amount of goods movement in an area, the raw materials and prefabricated components needed for the construction of green buildings in the area may not be sufficient, and they need to rely on transportation to meet the demand for goods. The high cost of transporting prefabricated components increases the input cost of green buildings, thus resulting in a decrease in overall efficiency.

(2) The traffic network density represents the traffic flexibility of a region [67]. The higher the road network density is, the more options there are for transporting materials, theoretically reducing the cost of transporting green buildings. However, maybe due to the high sensitivity of green buildings to transportation costs, the transportation costs of prefabricated components and raw materials are too high to offset the convenience brought about by transportation flexibility. Therefore, although the traffic network density has a positive impact on the efficiency of green buildings, it is not significant.

(3) It is usually smoother and more convenient for factor flowing in areas with high traffic network density, and the radius of economic activities also expands accordingly. Therefore, in the initial stage of green building development, such areas are more likely to

gather various elements to form a scale effect, which is the possible reason why the impact of traffic network density on SE is significantly higher than on PTE.

**Table 4.** The regression results of transportation infrastructure factors affecting green building efficiency in transportation cost model.

| Variable | Regression Results | | |
| --- | --- | --- | --- |
| | TE | PTE | SE |
| Core Variables | | | |
| Volume | −0.3118 * | −0.216 * | −0.1589 |
| | (−2.05) | (−2.52) | (−1.87) |
| Density | 0.2245 | 0.1001 | 0.1671 * |
| | (1.73) | (1.34) | (2.26) |
| Control Variables | | | |
| GDP | −0.2187 | −0.28 ** | 0.0606 |
| | (−1.38) | (−3.07) | (0.67) |
| Enterprise | −0.1321 | −0.155 * | −0.0807 |
| | (−2.48) | (−1.97) | (−1.04) |
| R&D | 0.1263 * | 0.0659 * | 0.0321 |
| | (2.23) | (2.02) | (0.99) |
| Employee | −0.18 | −0.0705 | −0.0659 |
| | (−0.87) | (−0.59) | (0.56) |
| Constant | 2.4 * | 4.237 *** | −0.0411 |
| | (2.22) | (7.14) | (−0.07) |
| Observations | 300 | 300 | 300 |
| Pseudo R$^2$ | 0.1 | 0.12 | 0.15 |

Note: t-statistics in parenthesis. *** $p < 0.001$, ** $p < 0.01$, * $p < 0.5$.

## 5. Discussion

### 5.1. Spatial Distribution of Green Buildings Efficiency

According to the efficiency value obtained from the DEA model, whether adjusted or not, the PTE of green buildings is significantly higher than SE, but TE is generally guided by SE whose trend is relatively consistent. Therefore, the key to improving the efficiency of green buildings is to improve SE. According to the regression results of the SFA model, the improvement of green space coverage rate and the local population's education level can significantly improve the efficiency of green buildings, as well as the PTE and SE. Therefore, local government departments could improve the efficiency of green buildings by increasing the green space coverage rate, increasing investment in local education or attracting highly educated talents. It can be found from Figure 3 that the spatial distribution of green building efficiency closer to the real situation is obviously uneven. The central government could also narrow the gap of nationwide green building efficiency by increasing subsidies to public green space or by rationally allocating educational resources in low-efficiency areas.

### 5.2. The Impact of Transportation Infrastructure on Green Building Efficiency in the Arrival Cost Model

In the arrival cost model, the regression results of fixed asset investment in municipal transportation infrastructure, road area per capita and number of stations per capita are all significantly positive, indicating that these three transportation infrastructure factors have a significantly positive impact on the efficiency of green buildings. However, in the neutralization analysis of TE, PTE and SE, only the number of stations per capita is always significantly positive. Therefore, when the government increases the funds for municipal transportation infrastructure and implements urban transportation planning, it should consider the reasonable allocation of public transportation stations. When planning the site selection of green buildings, the conditions of the surrounding transportation infrastructure should also be properly considered, and the input–output efficiency can be optimized by reducing the original value to attract developers and residents.

*5.3. The Impact of Transportation Infrastructure on Green Building Efficiency in the Cost of Transportation Model*

In the transportation cost model, the impact of freight volume on green building is significantly negative, while the impact of traffic network density on green building efficiency is positive but not significant. The same positive and negative relationship of regression coefficients is also reflected in the measurement results of PTE and SE. Because freight cost is more significant than transportation convenience, the key to reducing transportation cost is to reduce the amount of freight. A feasible measure is to increase the local production base of prefabricated components for green buildings and reduce the distance from the origin of prefabricated components to the construction site by increasing the number of manufacturers so as to achieve the purpose of reducing the transportation cost. A scientific and reasonable layout of the production base is beneficial not only to avoid an imbalance between supply and demand, but also to achieve sustainable development of green buildings.

## 6. Conclusions

This manuscript analyzes the input–output efficiency of green buildings in China and explores the degree of influence and direction of transportation infrastructure on the efficiency of green buildings. The innovations are as follows:

(1) The efficiency of green buildings that is closer to the real situation is calculated. This manuscript combines the super-efficiency SBM model with the three-stage DEA model to eliminate environmental factors and statistical noise. The research object is the green building efficiency of 30 provincial administrative regions in China from 2010 to 2020. It is more comprehensive than the existing research in both time and space, so the green building efficiency of China is closer to the real situation.

(2) Two paths of transportation infrastructure affecting the efficiency of green buildings were analyzed. This manuscript adopted the Tobit regression model to analyze the impact of transportation infrastructure on the efficiency of green buildings from the two paths of arrival cost within the city and transportation cost between regions, which are of academic and practical significance.

The main conclusions of this manuscript are as follows:

(1) Green space coverage and population education level have a significant positive impact on green building efficiency, which can be used to narrow the gap in green building efficiency nationwide. Compared with pure technical efficiency, scale efficiency plays a dominant role in overall technical efficiency, which is consistent with the current development law of green building. Well-developed transportation infrastructure system leads to the improvement of scale effect, and then technological progress gradually plays a leading role through knowledge spillover and industrial scale economy brought about by spatial agglomeration.

(2) In terms of geographical distribution, the efficiency of green buildings in China shows a trend of convergence from the eastern and western regions to the central regions, forming an obvious internal cluster effect, but the low efficiency areas are always concentrated in the northeast and southwest regions, which are less affected by radiation.

(3) Transportation infrastructure influences the efficiency of green buildings through two paths. In the analysis of arrival cost within the city, the investment in fixed assets of municipal transportation infrastructure, per capita road surface area and the number of public stations have a significant positive impact on the efficiency of green buildings. In the analysis of regional transportation cost, the impact of freight volume is significantly negative, and the impact of transportation network density on green building efficiency is positive but not significant.

(4) Overall, the impact of transportation infrastructure on the arrival cost of green buildings is more significant than the impact of transportation cost.

**Author Contributions:** Conceptualization, G.L.; methodology, G.L.; software, X.M.; validation, X.M.; formal analysis, X.M.; investigation, X.M.; resources, X.M.; data curation, X.M.; writing—original draft preparation, X.M.; writing—review and editing, G.L. and Y.S.; visualization, X.M.; supervision, Y.S.; project administration, Y.S.; funding acquisition, Y.S. All authors have read and agreed to the published version of the manuscript.

**Funding:** This research was funded by the National Natural Science Foundation of China (grant number 71872197).

**Institutional Review Board Statement:** Not applicable.

**Informed Consent Statement:** Not applicable.

**Data Availability Statement:** The data present in this study is available on request from the corresponding author.

**Conflicts of Interest:** The authors declare no conflict of interest.

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
