# Peer review of "Green Building Efficiency and Influencing Factors of Transportation Infrastructure in China: Based on Three-Stage Super-Efficiency SBM–DEA and Tobit Models"

_buildings, doi:10.3390/buildings12050623_

Round 1

Reviewer 1 Report

The research focuses the green building efficiency and its influence factors on transportation infrastructure in China. It shows traffic networks technology positive impact, while negative on the freight volume.

It adds the framed context to the subject area compared with other published material.  

Authors could consider improvements in onsite data gathering.

Further controls should be considered: Styrofoam models do not comply with the natural conditions; the reduced opens undermine airspeed (inertial to liquid behavior), inflating the temperature.  

The references are appropriate.  

Reviewer 2 Report

  1. I suggest that the authors provide the full name of all abbreviations (SBM, DEA, DMU…, etc.) when it is firstly shown in the texts.
  2. Line 45-49, Please provide the resources of energy consumption results for buildings (46.7%) and construction (28%).
  3. Please describe the meaning of Line 120-124 conditions related to formula (1) in Line 119. Also, it applies to formula (2) in Line 128 as well.
  4. Table 1, please describe the meaning of column “S.D.”; In addition, the unit of the first row: “Number of enterprises in the construction industry “ is shown as “unit”, please explain what it means? (ie. It is an individual company, a group, or an industrial type?) furthermore, the mean of all variables is strictly close to the minimum, is it normal?
  5. Please show the R square value of the regression results, which are shown in Table 3 and Table 4.

Reviewer 3 Report

The manuscript describes an analysis for the factors that influence the transportation infrastructure, and green building efficiency in and across China. This topic is certainly interesting, and the article is generally well-written. I would suggest, however, proof-reading some text, since some typos can be found here and there. After reviewing the paper, I recommended some corrections that address concerns I had over the method, both in terms or the overall structure and some minor, yet important details that I thought were missing. If the authors are willing to make these changes, this could help enhance the readership of their work. Some analytical suggestions have also been provided, which have been, along with the abovementioned, documented in more detailed comments below.

Introduction:

#1: P1, L30-34: Along with emission and energy challenges [ref. 2], the development of green buildings also seek to enhance human health, wellbeing and comfort (i.e. end-user experience), which are an equal mainstream challenges to the former in academic research (please see: Altomonte et al., 2019. Indoor environmental quality and occupant satisfaction in green-certified buildings. Building Research and Information). This should be acknowledged, since this also acts as barriers to sustainable design along with monetary constraints that were mentioned on line 40. I would also strongly urge the authors to provide a clear definition for what “green building” refers to, and which programmes or certification systems this may encompass.

#2: P2, L49-50: Please specify the green building programme (e.g. three star?), which has been developing over the last decade.

#3: P2, L61: While LEED is a popular and international recognised green certification system, it was not clear how this influences the built environment in the context of China (e.g. is it a widely implemented certification system in China, and what other green labels are considered?); however, if LEED was mentioned to highlight the importance of green buildings, I believe that other certification systems should also be given.

#4: P2, L68-70: Although these references are relevant to the study purview, I scope of research for green buildings is much wider than five factors the authors have listed. Reading further, I believe that these are relevant to research methods, and how data are analysed. If this is correct, please consider clarifying line 68 to reflect this fact.

Method:

#5: P2, L89: Please briefly explain the index and what it aims to achieve.

#6: Although the more intricate details for the method were generally well explained, the structure could have been better articulated. I would recommend that the section 3 (data and variables) is moved above the description for the analysis, and moved into section 2 (methods), since both the variables and data are part of the method. Unless I missed it, please also specify at the beginning where the data was sourced or derived from. Reading further into the method, the datasets are explained on page 8. Please consider moving section 3.5 at the start of the method. Most importantly, please give more details that explain characteristics for the data. For example, how much data was contained, and how much was collected per year. Table 1 provides some perspective on this, but the values are aggregated.

#7: Table 1: The information in the table is not clear. The number of enterprises and green buildings should be integer (i.e. count), and should not be taken as a mean average. If this is aggregated across different conditional variables, descriptive statistics for each can be provided.

#8: P7, L255-263: Many statements are not entirely clear here, and I would recommend revising this text also for typos (e.g. “have a higher tendency to green consumption”). It is also not clear what “more correct understanding”, “green consumption”, “green products”, and “green life” refer to. Please consider amending this accordingly.

#9: P8, L288: Please state how much data was collected within this timeframe, and also whether any major changes occurred within the design certification standard (e.g. assignment of certain credits) that may have influenced the outcome of the study (e.g. it is possible that some design recommendations were proposed later during the development of the standard). Dataset size becomes vital to the interpretation of the results presented in Tables 2, 3 and 4, which rely solely on statistical significance, giving no real indication for the size of relationships analysed, besides the regression coefficients.

Results:

#10: Tables 2, 3 and 4: Please state whether the values are the regression coefficients in the table captions. Further, why are statistically significant results denoted by p-values that are less than 10% (p<0.10)? This is not a conventional threshold. My understanding is that thresholds are typically denoted by values that are less than: 0.05*, 0.01** and 0.001***. Please also check: “represent significant at” -> “represent statistically significant coefficients less than”. If possible, the r2 (or equivalent) could also be given to help gauge the size of the relationships that are described.

#11: P8, L308-317: The interpretations given to the regression models are not very accurate. Statistically significant results denote models with predictors that can predict the outcome variable. I would also suggest saying the variables are positively or negatively related, rather than correlated, since the authors did not opt to use correlation analyses.

#12: Figure 3: Please consider grouping common scales and legends, and enlarging them above the figure images. They are currently too small, and the text inside the map cannot be seen. Also, the antonym for “Insufficient” should be “Sufficient” and not “High Efficient”, which should be “Low Efficient”.

Reviewer 4 Report

The paper presents the results of a three-stage super-efficiency SBM DEA and Tobit model to locate the factors of the transportation infrastructure in Chine that influence green building efficiency.

The manuscript is well written and documented.

I have only a few comments:

  • The authors focus more on the documentation and presentation of the methodology approach rather than on the discussion of the results. I miss the section on the Discussion of your findings in the light of previous literature, which is a significant part of the article.
  • If you add the aim of the application of each step of the Methodology approach in Figure 1, you would make it readable.
  • What new in the research does your methodology bring? You could add a few sentences in the Conclusions section.

Round 2

Reviewer 4 Report

The authors respond to the reviewers' comments satisfactorily.